# Short-Term Mortality in Patients with Heart Failure at the End-of-Life Stages: Hades Study

**DOI:** 10.3390/jcm11092280

**Published:** 2022-04-19

**Authors:** Miguel Angel Muñoz, Esther Calero, Julio Duran, Elena Navas, Susana Alonso, Nuria Argemí, Marta Casademunt, Patricia Furió, Elena Casajuana, Nuria Torralba, Nuria Farre, Rosa Abellana, José-Maria Verdú-Rotellar

**Affiliations:** 1Gerencia Territorial de Barcelona (Primary Healthcare), Institut Català de la Salut, 08007 Barcelona, Spain; salonso.bcn.ics@gencat.cat (S.A.); nargemi.bcn.ics@gencat.cat (N.A.); mcasademunt.bcn.ics@gencat.cat (M.C.); pfurio.bcn.ics@gencat.cat (P.F.); ecasajuanaan.bcn.ics@gencat.cat (E.C.); ntorralba.ics@gencat.cat (N.T.); verdujm@gmail.com (J.-M.V.-R.); 2Departament de Ciències Experimentals i de la Salut, School of Medicine, Universitat Pompeu Fabra, 08002 Barcelona, Spain; 3Institut Universitari d’Investigació en Atenció Primària Jordi Gol (IDIAP Jordi Gol), 08007 Barcelona, Spain; enavas@idiapjgol.info; 4Bellvitge University Hospital, Institut Català de la Salut, 08921 Barcelona, Spain; ecalero.bcn.ics@gencat.cat; 5Clinica Sant Antoni (Institut Medic i de Rehabilitació), 08038 Barcelona, Spain; jduran@csantantoni.com; 6Hospital del Mar Medical Research Institute, 08003 Barcelona, Spain; nfarrelopez@psmar.cat; 7Departament de Fonaments Clínics-Bioestadística, School of Medicine, Universitat de Barcelona, 08007 Barcelona, Spain; rabellana@ub.edu

**Keywords:** heart failure, community setting, predictive model, mortality, short term prognosis, end of life, prognostic

## Abstract

Background: Information regarding short-term vital prognosis in patients with heart failure at advanced stages of the disease is scarce. Objective: To develop a three-month mortality predictive model for patients with advanced heart failure. Methods: Prospective observational study carried out in primary care and a convalescence community facility. Heart failure patients either New York Heart Association (NYHA) III with at least two HF hospitalizations during the previous six months or NYHA IV with/without previous recent hospitalization were included in the study. Multivariable predictive models using Cox regression were performed. Results: Of 271 patients included, 55 (20.3%) died during the first three months of follow-up. Mean age was 84.2 years (SD 8.3) and 59.8% were women. Predictive model including NT-proBNP had a C-index of 0.78 (95% CI 0.71; 0.85) and identified male gender, low body mass index, high potassium and NT-proBNP levels, and moderate-to-severe dependence for daily living activities (Barthel index < 40) as risk factors of mortality. In the model without NT-proBNP, C index was 0.72 (95% CI 0.64; 0.79) and, in addition to gender, body mass index, low Barthel index, and severe reductions in glomerular filtration rate showed the highest predictive hazard ratios for short-term mortality. Conclusions: In addition to age, male gender, potassium levels, low body mass index, and low glomerular filtration, dependence for activities of daily living add strong power to predict mortality at three months in patients with advanced heart failure.

## 1. Introduction

Heart failure (HF) is a progressive condition which follows a trajectory of organ failure. It implies a gradual decline, presenting intermittent episodes of exacerbation, leading to death, which is generally unpredictable [1,2], although almost 50% of patients die within 5 years of diagnosis [3].

Many scores and risk functions have been developed to predict HF prognosis. Nevertheless, most models are limited to individuals with reduced left ventricular ejection fraction (LVEF), are used to determine long-term outcomes, or are focused only on hospitalized patients and selected populations included in randomized clinical trials [4,5,6,7,8,9]. The majority of prognostic risk tools for HF do not report clinically relevant thresholds for sensitivity and specificity. As a consequence, current guidelines encourage investigating accurate prognostic risk tools in predicting HF mortality [10].

Moreover, current risk scores do not usually include prevalent conditions with prognostic impact, and evidence about survival in HF patients attended to in community settings is still scarce and inconsistent [11,12].

Most elderly patients with HF attended to in primary care are not candidates for advanced therapies such as cardiac transplantations and left ventricular assisted devices [13]. In addition, although survival of such patients is often comparable to those with cancer, there are no reliable tools to predict short-term mortality and include these individuals in palliative care programs [14].

This study aims at developing a three-month mortality predictive model which permits healthcare professionals, patients and families to better approach the end-of-life care of elderly patients with advanced HF.

## 2. Methods

### 2.1. Design

The HADES study (Heart failure at ADvancEd Stages) was a prospective, multicentric cohort study aimed at developing a predictive model of mortality in patients at advanced stages of HF.

Recruitment was from June 2017 to December 2019. Follow-up continued until December 2020. We are presenting results from the entire cohort.

### 2.2. Participants

We included patients older than 40 years with stable HF. They were required to be either New York Heart Association (NYHA) III and to have had at least two HF hospitalizations during the previous six months, or be NYHA IV with/without previous recent hospitalization.

Individuals with terminal cancer, candidates for heart transplantation, and subjects presenting severe mental illness were excluded from the study.

Patients were identified from primary care electronic medical records, from the discharge report of hospitals in Barcelona, and from a community-based institution which offers convalescence care and temporal support to HF patients transferred from acute care hospitals (Clinica Sant Antoni).

The study was coordinated from a primary healthcare research unit in Barcelona. Information was collected by 27 nurses providing the usual care to HF patients in 22 primary healthcare centers, 2 university hospitals (Bellvitge Hospital and Hospital del Mar), 1 acute care hospital (ESIC Dos de Maig), and the community-based Clinica Sant Antoni.

The endpoint was taken to be all-cause mortality occurring during the three months after study inclusion. Mortality was assessed by consulting medical records and health administrative registries and, in case of doubt, the patient’s relatives. All-cause and specific mortality causes were registered.

Social and demographic variables were gathered from the patients through a questionnaire. Functional status was assessed by the Barthel index, previously validated for use in Spanish speaking populations [15]. This index is widely employed to assess independence in carrying out activities of daily living such as personal care and mobility.

Clinical variables including body mass index (BMI), blood pressure, heart rate, and respiratory frequency were measured by clinical examination.

Blood samples were also collected when no recent information was available (one week before).

Based on the results of a previous study [16], and to permit ease of use of the prognostic model for clinicians, BMI was split into two categories (<25 kg/m^2^ and above).

As N-terminal pro-B-type natriuretic peptide (NT-proBNP) showed high variability, it was divided into quartiles to facilitate analysis. Since many countries do not have access to natriuretic peptides in primary care settings, we developed two predictive models, one of them including NT-proBNP and other without. Finally, the Barthel index was grouped into two categories, a score < 40 representing severe dependence (0 = totally dependent and 100 = completely independent) [17].

Information about comorbidities, year of HF diagnosis, LVEF, and medication was obtained from medical records.

### 2.3. Statistical Analysis

Descriptive statistics were used to summarize general data. Continuous variables are expressed by mean and standard deviation (SD) or medians and interquartile range, and the categorical variables by frequencies and percentages.

The significant association of all variables with a three-month prognosis of mortality was evaluated using the univariate Cox model. A *p* value < 0.05 was required to include candidate variables in the subsequent stepwise selection procedure.

A multivariate Cox model was used to identify predictive factors. The final model was determined based on Akaike Information Criteria (AIC).

We estimated the discrimination and calibration of the model to analyze its performance. Internal validation was carried out using bootstrap resampling. This method is based on generating new samples of the same size as the original sample by replacement sampling. We generated 100 random samples from the original data set.

To determine the discrimination capacity of the model, Harrell’s C statistic (concordance index) was used, and to evaluate calibration we established a graph comparing the predicted probabilities in the model versus the observed ones.

### 2.4. Scoring System

In order to build a score to predict the risk of death at three months, we calculate how far each category of each risk factor is from the base category in terms of regression units using β_i (W_ij-W_iREF). We define the constant “B” for the point system. Here we let “B” reflect the increased risk associated with an increase in age of 10 years. We determine the points associated with each category of each risk factor using Points_ij = β_i (W_ij-W_iREF)/B and rounded to the nearest integer.

Using statistical criteria, we divide the score into quartiles to define low risk, intermediate risk, high risk, and very high risk.

### 2.5. Risk Score in the Model with NT-ProBNP

The scores range from 0 to 27 points. Reference risk factor (0 points) corresponds to the profile of a woman, with a body mass index greater than or equal to 25 kg/m^2^, potassium levels less than or equal to 5 mEq, glomerular filtration rate with values greater than or equal to 30 mL/min, NT-proBNP values less than or equal to 1646 and Barthel index ≥ 40.

According to risk groups, the low risk of mortality at three months corresponds to score values less than 4, intermediate risk to score values equal to or greater than 4 and less than 7, high risk values are equal to or greater than 7 and less than 10, and finally, very high risk values are equal to or greater than 10. Therefore, the mean probability of mortality at three months in the low risk group is 3.80%, in the intermediate risk group 9.91%, for high risk 20.8%, and for the very high risk group the mean probability is 49.3%. In the study cohort, the cumulative incidence for low risk patients was 1.75%, for intermediate risk 2.48%, for high risk 9.64%, and for very high risk 21.63%.

### 2.6. Risk Score in the Model without NT-ProBNP

In this case, the scores range from 0 to 27 points. The reference point in the score system corresponds to the profile of a woman with a body mass index greater than or equal to 25 kg/m^2^, potassium levels equal to or less than at 5 mEq, glomerular filtration rate greater than or equal to 30 mL/min and a Barthel index ≥ 40.

The risk groups were defined as follows: score values below 4 were considered low risk, intermediate risk score values were equal to or greater than 4 and less than 6, high risk values were equal to or greater than 6 and less than 8, and very high risk values were equal to or greater than 8. The mean probability of mortality at three months for the low, intermediate, high, and very high risk groups was 6.91%, 13.7%, 22.3%, and 45.5% respectively. The cumulative incidence for the low risk is 3.28%, for intermediate risk 4.66%, high risk 7.20%, and very high risk 21.25%.

Statistical analysis was conducted using R Software for Windows version 3.6.1, Vienna, Austria.

## 3. Results

The HADES study included 276 NYHA III/IV patients. Among those initially selected, three were excluded due to not meeting inclusion criteria, and two declined to participate.

Of the 271 participants finally analyzed, 55 (20.3%) died during the first three months of follow-up, which represented a 40.7% mortality during the first year. In 38 of these patients (70.4%), death was due to HF complications.

The patients’ baseline characteristics are summarized in Table 1. Mean age was 84.2 years (SD 8.3) (53.0% > 85 years and 87.4% > 74 years) and 59.8% were women. Most patients presented preserved ejection fraction (69%), with a mean LVEF of 53.4% (SD 14.5). The majority were NHYA III and had been hospitalized in the previous six months (183, 67.5%); 88 (32.5%) patients were NYHA IV at study inclusion.

Univariate analysis showed that male gender, older age, worse glomerular filtration, and increased levels of potassium, creatinine, and NT-proBNP were associated with a greater probability of dying. In contrast, higher BMI, systolic blood pressure, and Barthel index were protective.

No effects on mortality were found for LVEF, heart rate, or any of the analyzed comorbidities.

### 3.1. Predictive Model with NT-ProBNP (Model 1)

The multivariate predictive model showed that male gender, low BMI, high potassium and NT-proBNP levels, and moderate-to-severe dependence for daily living activities (Barthel index < 40) predicted a greater risk of dying (Table 2).

The discrimination capacity of the model (C-index) was 0.78 (95% CI 0.71; 0.85).

### 3.2. Predictive Model without NT-ProBNP (Model 2)

Since many countries do not have access to NT-proBNP in primary care settings, we calculated a second predictive model removing this variable. The discrimination index was slightly lower (C index = 0.72, 95% CI 0.64; 0.79).

Male gender, BMI, and the Barthel index (<40) remained in the model and were associated with greater risk. We also found that severe reductions in glomerular filtration rate showed the highest predictive hazard ratios for short-term mortality.

### 3.3. Validation of the Predictive Models

We internally validated the models by bootstrapping, generating one hundred different samples. The validation demonstrated a discrimination index of 0.73 for the predictive model including NT-proBNP and 0.67 for the model without.

The agreement between observed outcomes and model predictions for the two models (calibration) is represented in Figure 1.

### 3.4. Risk Score in the Models

In order to build a score able to predict the risk of mortality at three months we applied the methodology proposed by Sullivan et al. [18].

### 3.5. Risk Score in the Model with NT-ProBNP

We created three risk groups based on the application of tertiles for the probability of mortality. A <8% probability of mortality at 3 months indicated low risk, 8–20% intermediate, and >20% high. Thus, the low risk of mortality at 3 months corresponded to scores ≤ 4, intermediate 5–7, and high ≥ 8. In our population, the cumulative incidence for low-risk patients was 1.3%, 4.5% for intermediate, and 17.9% for high.

The score ranks from 0 to 21. A score of 0 corresponds to female gender, age 55 to 64 years, BMI ≥ 25 kg/m^2^, potassium value ≤ 5 mmol/L, glomerular filtration rate ≥ 30 mL/min, NT-proBNP values ≤ 1646 pg/mL, and Barthel index ≥ 40. The estimated risk for 0 points was 0.020 (Figure 2A).

### 3.6. Risk Score in the Model without NT-proBNP

The score ranged from 0 to 15 points. A score of 0 corresponds to female gender, age 55 to 64 years, BMI ≥ 25 kg/m^2^, potassium value ≤ 5 mmol/L, glomerular filtration ≥ 30 mL/min, and Barthel index ≥ 40.

We created three risk groups based on the application of tertiles for the probability of mortality. A <10% probability of mortality at 3 months indicated low risk, 10–20% intermediate, and >20% high. Thus, the low risk of mortality at 3 months corresponded to scores ≤ 4, intermediate 4–6, and high ≥ 7. In our population, the cumulative incidence for low-risk patients was 3.6%, 4.1% for intermediate, and 15.6% for high (Figure 2B).

## 4. Discussion

The HADES study identified four independent variables which accurately predict short-term (three months) mortality in patients with advanced HF.

Male gender, low BMI, dependence for daily living activities, and high levels of natriuretic peptides and potassium were found to be associated with this excess of mortality. The discrimination (ability to separate individuals who develop events from those who do not) was above 0.70, and the calibration (agreement between the estimated and the “true” risk of an outcome) showed good concordance.

Many models have been developed in the previous two decades to predict mortality in HF patients, but none have been specifically developed in a population of elderly patients with advanced HF [4,5,6,7,8,9].

A recent systematic review analyzed 40 risk-prediction publications regarding HF prognosis. Nine studies, however, focused on cardiovascular mortality rather than all-cause mortality, and only one performed internal validation. Moreover, most of them were developed to calculate mortality at one or two years, which is not applicable to a high-risk population such as ours [19].

A recent paper published by our group described some logistic models to explore the variables related to higher mortality during the first year of reaching NYHA IV. It was, however, based on electronic medical records [16].

Prediction models determined from administrative datasets may lead to bias in the real clinical world [20]. We, therefore, decided to carry out this prospective study, measuring all the variables by evaluating every individual included in the follow-up.

Regarding predictors, the most frequently identified variables in previous publications have been age, male gender, NT-proBNP, systolic blood pressure, and renal function, by analyzing blood urine nitrogen and creatinine [19].

It has been reported that for patients admitted to hospital emergency rooms for acute HF, mortality at 30 days, three months, and one year was higher when dependency for daily activities was severe [21,22,23]. Our participants, however, were stable patients in the community or convalescence units. Thus, a major contribution of our model is the relevance of disability to such individuals’ vital progress.

Natriuretic peptides are cleared by the kidneys. In the case of renal failure leading to hypervolemia and hypertension, the secretion of these biomarkers increases [24]. NT-proBNP, released during hemodynamic stress, has been reported to be an independent predictor of mortality in patients presenting acute [25] and stable heart failure [26], a finding which concurs with our results.

Kenchaiah et al. described a 45.7% mortality excess in HF patients with a BMI 22.5–24.9, and a rise of up to 69.4% in those with a BMI < 22.5 kg/m^2^ [27]. Such an association has been explained by cardiac cachexia, occurring from a combination of inadequate protein intake, nutrient malabsorption, catabolic processes, and inadequate anabolism [28]. This condition is present in 61% of NYHA III-IV hospitalized patients [29].

Advanced NYHA implies a deterioration in the patient’s functional status, hindering any physical effort. It has been identified as a prognostic factor for hospitalization by some authors such as Formiga et al., Shudakar et al., and Upshaw et al. [30,31,32], whilst others assessed mortality including Freudenberger et al., Barlera et al. [33,34], the MAGGIC study [5], and Levy et al. [6]. In these studies, however, the follow-up was as long as one year.

The EMPHASIS-HF trial found that impaired renal function and hyperkalemia were interrelated and associated with worse prognosis in HF patients [35]. Núñez et al. observed that elevated serum potassium levels, mostly related to comorbidities and HF medication, were independently associated with mortality in such patients [36].

Our predictive model was developed in a cohort of advanced age patients, most of whom were attended to in a primary care setting. It may help clinicians forecast what kind of patient will present the worst short-term prognosis. Knowing the probability of short-term survival could help plan a better care proposal for patients and families at the terminal stages of the disease. Offering an adequate palliative approach is crucial not only to alleviate symptoms, but also to manage the anxiety and fear of patients. It has been described that patients able to establish advanced care planning were more satisfied with their end-of-life support [37].

It is crucial to prescribe therapies to alleviate symptoms and anxiety at the various stages of the disease, as well as those treatments oriented to improve daily activities. However, health professionals should also to take into account the possibility of avoiding medication which does not improve lifespan and may even affect the patient’s quality of life.

### Strengths and Limitations

In spite of the high mortality in patients with advanced HF, the percentage of events at three months in our sample may have been insufficient to detect statistical differences in some of the variables, such as glomerular filtration in the first model.

We had 10% and 13% missing values in NT-proBNP and ejection fraction, respectively. These figures are low when considering that most of patients received domiciliary care and could not easily access an echocardiogram.

One of the goals of our study was to find an accurate predictive model by using variables easily obtained at the patient’s bedside. Since NT-proBNP is not routinely determined in a primary care setting, we calculated an alternative predictive model which showed a good performance.

Some strengths of our study were the prospective inclusion of a considerable number of very elderly patients, the focus on advanced HF, the evaluation of dependence in daily living activities in patients at the community level, and the high percentage of patients with preserved ejection fraction.

Every patient is different, and conversations between healthcare professionals and patients about end-of-life care should take place regularly along the disease trajectory and not rely solely on a score.

In order to implement palliative care, it is necessary to take into account the differences in population-based patterns and clusters at the end of life [38]. Our proposal is just to help clinicians and patients to take decisions.

More studies will be necessary to determine if the score may affect not only patients’ prognosis but also drugs and device prescription.

## 5. Conclusions

An accurate predictive model based on variables easily obtained at the community level may help clinicians to properly approach end-of-life care in elderly patients with heart failure. In addition to age, male gender, potassium levels, low body mass index, and glomerular filtration, a moderate–severe dependence for activities of daily living add strong power to predict mortality at three months in patients with advanced heart failure.

## Figures and Tables

**Figure 1 jcm-11-02280-f001:**
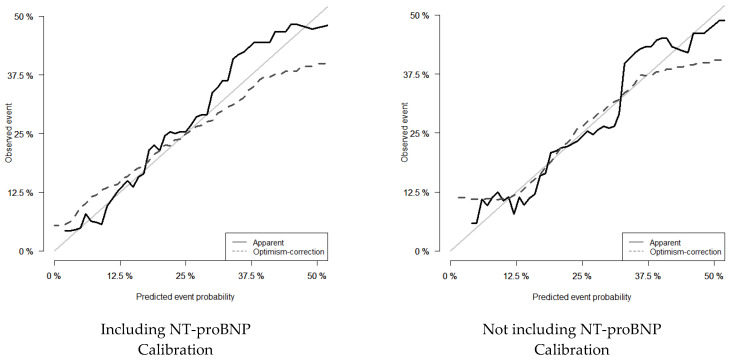
Observed versus expected model-predicted 3-month mortality in patients with advanced heart failure.

**Figure 2 jcm-11-02280-f002:**
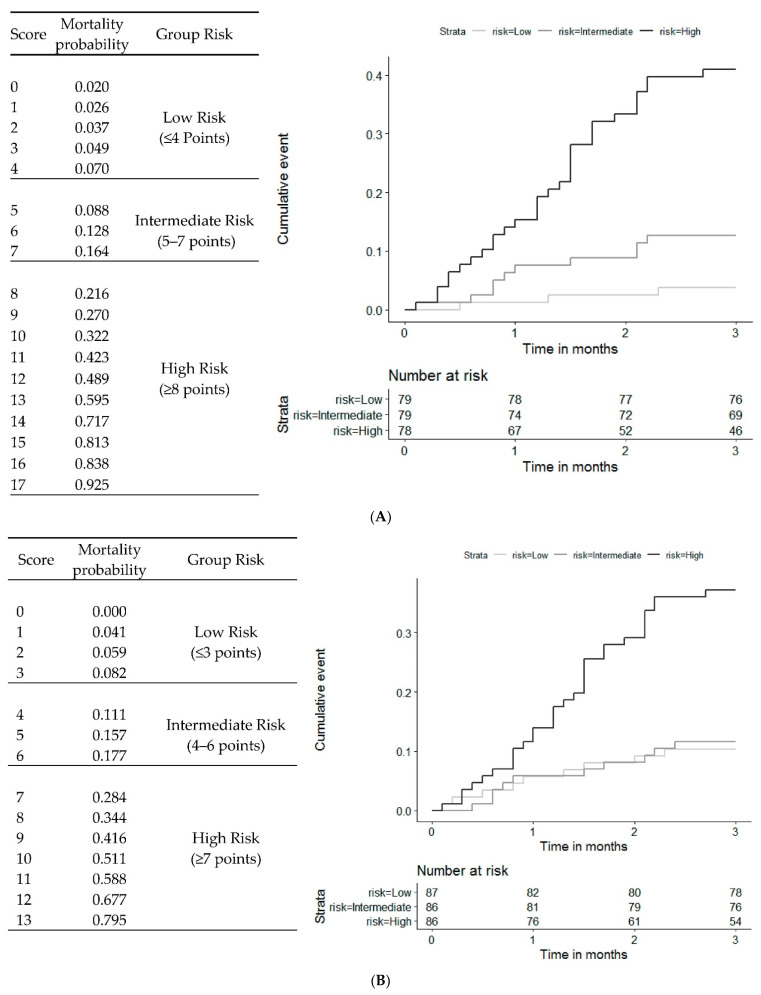
Three-month mortality according to the score risk group. (**A**): Model with NT-proBNP. (**B**): Model without NT-proBNP.

**Table 1 jcm-11-02280-t001:** Univariate analysis of variables related to short-term (three months) mortality in patients at NYHA III-IV.

	All *n* = 271	Alive *n* = 216	Dead *n* = 55	Hazard Ratio (95% Confidence Interval)	*p* Value
Women (%)	162 (59.8)	135 (62.5)	27 (49.1)	Ref.	
Men (%)	109 (40.2)	81 (37.5)	28 (50.9)	1.59 [0.94; 2.70]	0.084
Age (Mean, SD)	84.2 (8.3)	83.6 (8.5)	86.5 (7.2)	1.04 [1.01; 1.08]	0.019
Age groups (years) (%)					
<65	6 (2.2)	6 (2.7)	0 (0.0)	0.00 [0.00; 0.00]	
65–74	28 (10.3)	23 (10.6)	5 (9.1)	Ref:	
75–84	93 (34.3)	80 (37.0)	13 (23.6)	0.76 [0.27; 2.12]	0.596
>84	144 (53.1)	107 (49.5)	37 (67.3)	1.52 [0.60; 3.87]	0.380
**Level of Studies (%)**					
Less than secondary education	199 (73.7)	161 (74.9)	38 (69.1)	Ref.	
Secondary school	55 (20.4)	41 (19.1)	14 (25.5)	1.39 [0.75; 2.57]	0.289
University studies	16 (5.9)	13 (6.0)	3 (5.4)	0.92 [0.28; 2.98]	0.890
**Clinical variables** (Mean, SD)					
Left ventricular ejection fraction	53.4 (14.6)	53.8 (14.8)	51.7 (13.6)	0.99 [0.97; 1.01]	0.412
Body mass index (kg/m^2^)	28.0 (5.7)	28.4 (5.9)	26.0 (4.5)	0.93 [0.88; 0.98]	0.006
Systolic blood pressure (mmHg)	121 (19.0)	122 (18.9)	117 (19.3)	0.99 [0.97; 1.00]	0.074
Diastolic blood pressure (mmHg)	66.0 (10.1)	66.6 (10.1)	63.9 (9.6)	0.98 [0.95; 1.00]	0.087
Heart rate (beats per minute)	76.0 (14.1)	75.8 (14.0)	77.0 (14.5)	1.01 [0.99; 1.02]	0.528
Respiratory frequency (per minute)	19.5 (3.4)	19.4 (3.5)	19.7 (2.9)	1.02 [0.94; 1.10]	0.642
**Laboratory variables** (Mean, SD)					
Sodium (mmol/L)	140 (6.8)	140 (7.4)	140 (3.6)	0.99 [0.96; 1.03]	0.775
Potassium (mmol/L)	4.3 (0.6)	4.2 (0.5)	4.4 (0.7)	1.67 [1.09; 2.55]	0.018
Creatinine (mg/dL)	1.4 (0.7)	1.3 (0.7)	1.6 (0.8)	1.47 [1.10; 1.96]	0.009
Glomerular filtration (mL/min)	47.5 (20.7)	49.0 (20.0)	41.9 (22.4)	0.98 [0.97; 1.00]	0.019
Hemoglobin (gr/dL)	11.4 (1.7)	11.5 (1.7)	11.0 (1.6)	0.86 [0.74; 1.01]	0.070
NT-proBNP (pcg/mL) (Median; P25–75)	3437 (1646; 8010)	3040 [1512; 5996]	7743 [3415; 16,080]	1.00 [1.00; 1.00]	<0.001
**Functional status**					
New York Heart Association (NYHA) (%)					
NYHA III	183 (67.5)	150 (69.4)	33 (60.0)	Ref.	
NYHA IV	88 (32.5)	66 (30.6)	22 (40.0)	1.45 [0.84; 2.48]	0.181
Barthel index (Mean, SD)	60.8 (25.3)	63.3 (24.5)	51.0 (26.2)	0.98 [0.97; 0.99]	0.001
**Comorbidity** (%)					
Current smoker	13 (4.8)	11 (5.1)	2 (3.6)	0.90 [0.21; 3.76]	0.881
Coronary heart disease	70 (25.8)	55 (25.5)	15 (27.3)	1.07 [0.59; 1.95]	0.812
Stroke	32 (11.8)	23 (10.6)	9 (16.4)	1.53 [0.75; 3.12]	0.246
Atrial fibrillation	134 (49.4)	106 (49.1)	28 (50.9)	1.05 [0.62; 1.78]	0.854
Hypertension	183 (67.5)	151 (69.9)	32 (58.2)	0.63 [0.37; 1.08]	0.090
Diabetes	100 (36.9)	80 (37.0)	20 (36.4)	0.98 [0.57; 1.70]	0.952
Chronic obstructive pulmonary disease	82 (30.3)	65 (30.1)	17 (30.9)	1.00 [0.57; 1.77]	0.997
Chronic kidney disease	109 (40.2)	85 (39.4)	24 (43.6)	1.16 [0.68; 1.98]	0.580
Anemia	109 (40.2)	86 (39.8)	23 (41.8)	1.07 [0.63; 1.83]	0.799
Depression	134 (49.4)	106 (49.1)	28 (50.9)	1.05 [0.62; 1.78]	0.854
**Medication**	*n* (%)	*n* (%)	*n* (%)		
Inhibitors of the renin-angiotensin system *	138 (50.9)	116 (53.7)	22 (40.0)	0.60 (0.35; 1.04)	0.067
Beta blockers	141 (52.0)	114 (52.8)	27 (49.1)	0.87 (0.51; 1.47)	0.603
Loop diuretics	214 (79.0)	173 (80.1)	41 (74.5)	0.75 (0.41; 1.38)	0.363
Mineral corticoid receptor antagonists	62 (22.9)	47 (21.8)	15 (27.3)	1.29 (0.71; 2.34)	0.400
Neprilysin inhibidors	11 (4.0)	8 (3.7)	3 (5.4)	1.26 (0.39; 4.05)	0.694
Ivabradin	4 (1.48)	4 (1.8)	0 (0.0)	-	-
Digoxin	26 (9.6)	19 (8.8)	7 (12.7)	1.43 (0.65; 3.16)	0.379
Statins	104 (38.4)	82 (38.0)	22 (40.0)	1.07 (0.63; 1.84)	0.799

Categorical variables are shown with *n* (%); continuous variables are shown with means and standard deviation; * Angiotensin-converting enzyme inhibitors or angiotensin-receptor blockers.

**Table 2 jcm-11-02280-t002:** Predictive model for three-month mortality in patients with advanced heart failure with N-terminal pro-B-type natriuretic peptide (NT-proBNP) and without NT-proBNP.

**Multivariate Model 1 (with NT-ProBNP)**					**Risk Score**	
**Variables**	**Categories**	** *n* **	**HR (95% CI)**	***p*-Value**	**Categories**	**Points**
Sex	Women	143	Reference		Women	0
	Men	93	2.64 (1.37; 5.09)	0.004	Men	3
Age	(years)	236	1.03 (0.99; 1.07)	0.147	55–64	0
					65–74	1
					75–84	2
					85–94	3
					94–104	4
Body mass index	≥25 kg/m^2^	166	Reference		≥25 kg/m^2^	0
	<25 kg/m^2^	70	1.90 (1.04; 3.49)	0.038	<25 kg/m^2^	2
Potassium(mmol/L)		236	1.54 (1.02; 2.33)	0.041	≤5 mmol/L	0
					>5 mmol/L	3
Glomerular filtration rate	≥30 mL/min	184	Reference		≥30 mL/min	0
	<30 mL/min	52	1.91 (0.99; 3.69)	0.055	<30 mL/min	2
NT-proBNP (pg/mL)	≤1646	61	Reference		≤1646	0
	1647–3437	59	0.75 (0.25; 2.37)	0.623	1647–3437	−1
	3438–8010	58	1.32 (0.50; 3.46)	0.570	3438–8010	1
	>8010	58	2.71 (1.09; 6.71)	0.032	>8010	3
Barthel index	≥40	195	Reference		≥40	0
	<40	41	3.04 (1.59; 5.83)	<0.001	<40	4
**Multivariate Model 2 (without NT-ProBNP)**					**Risk Score**	
**Variables**	**Categories**	** *n* **	**HR (95% CI)**	***p*-Value**	**Categories**	**Points**
Sex	Women	155	Reference		Women	0
	Man	104	1.93 (1.09; 3.43)	0.024	Man	2
Age	(years)	259	1.03 (1.00; 1.07)	0.071	55–64	0
					65–74	1
					75–84	2
					85–94	3
					94–104	4
Body mass index	≥25 kg/m^2^	182	Reference		≥25 kg/m^2^	0
	<25 kg/m^2^	77	2.09 (1.19; 3.68)	0.010	<25 kg/m^2^	2
Potassium (mmol/L)		259	1.43 (0.95; 2.16)	0.088	≤5 mmol/L	0
					>5 mmol/L	2
Glomerular filtration rate	≥30 mL/min	201	Reference		≥30 mL/min	0
	<30 mL/min	58	2.53 (1.42; 4.50)	0.002	<30 mL/min	3
Barthel index	≥40	211	Reference		≥40	0
	<40	48	2.30 (1.25; 4.22)	0.007	<40	2

## Data Availability

The data of this study are available from the IDIAP Jordi Gol Institute but restrictions apply to the availability of these data, which are not publicly available. Data may be, however, available from the authors upon reasonable request and with permission of the IDIAP Jordi Gol Institute.

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
