# Peer review of "Short-Term Mortality in Patients with Heart Failure at the End-of-Life Stages: Hades Study"

_jcm, 2022, doi:10.3390/jcm11092280_

Round 1
Reviewer 1 Report
The paper is well written and results are presented in a very interesting way. In particular the idea to set up a composite score is quiet valuable and very promising. However, some concerns may be raiser about the clinical impact of such evaluation in end stage HF. More studies are needed to determine if the score may affect not only patients' prognosis but could also impact drugs and device prescription.
Author Response
We sincerely thank the work made by the reviewers. We have revised the manuscript and we have added all the comments and suggestions proposed.
REVIEWER 1:
The paper is well written and results are presented in a very interesting way. In particular the idea to set up a composite score is quiet valuable and very promising. However, some concerns may be raiser about the clinical impact of such evaluation in end stage HF. More studies are needed to determine if the score may affect not only patients' prognosis but could also impact drugs and device prescription.
ANSWER:
We totally agree with the reviewer and we have added this comment to the strength and limitations section.
Reviewer 2 Report
In this article" SHORT-TERM MORTALITY IN PATIENTS WITH HEART FAILURE AT THE END-OF-LIFE STAGES: HADES Study" the authors developed a three-month mortality predictive model for patients with advanced HF. The authors identified male gender, low Body Mass Index, high potassium and NT-Pro BNP levels, and Barthel in-dex <40 as a risk factors of mortality for HF patients. Did the authors measure other cardiac biomarkers (hs-cTnI, sST2) besides NT-proBNP?
In general the work is well presented, however I would suggest to the authors to improve the caption of figures and tables to facilitate the reading of the results.
Author Response
We sincerely thank the work made by the reviewer. We have revised the manuscript and we have added all the comments and suggestions proposed.
REVIEWER 2
In this article" SHORT-TERM MORTALITY IN PATIENTS WITH HEART FAILURE AT THE END-OF-LIFE STAGES: HADES Study" the authors developed a three-month mortality predictive model for patients with advanced HF. The authors identified male gender, low Body Mass Index, high potassium and NT-Pro BNP levels, and Barthel in-dex <40 as a risk factors of mortality for HF patients. Did the authors measure other cardiac biomarkers (hs-cTnI, sST2) besides NT-proBNP?
ANSWER:
Dear reviewer, unfortunately we did not have access to other biomarkers to be determined for this project.
REVIEWER 2
In general the work is well presented, however I would suggest to the authors to improve the caption of figures and tables to facilitate the reading of the results.
ANSWER:
Following the recommendations of the reviewer we have revised the captions of the tables and figures.
Reviewer 3 Report
The paper introduces the new model which can be used to determine prognosis in patients with heart failure at advanced stages. Dependence during the activity of daily living was shown to add power to the prediction model applied to patients with advanced heart failure. It is well written, data support and conclusions, and the results are well illustrated in the figures and tables. As the minor point of improvement:
- Improved organization of the figure 2 which is now extending over two pages.
- Better alignment of the subtitles on the Figure 2.
- In the section "Risk score in the model with NTproBNP" in the methods section 1,75 % should be 1.75 % and 2,48 % should be 2.48 %
- Use of spaces between the number and following units should be controlled through the text of the manuscript.
- In the discussion relevance of the therapy aimed at the support of independent daily activities can be emphasized.
Author Response
We sincerely thank the work made by the reviewer. We have revised the manuscript and we have added all the comments and suggestions proposed.
REVIEWER 3
The paper introduces the new model which can be used to determine prognosis in patients with heart failure at advanced stages. Dependence during the activity of daily living was shown to add power to the prediction model applied to patients with advanced heart failure. It is well written, data support and conclusions, and the results are well illustrated in the figures and tables. As the minor point of improvement:
- Improved organization of the figure 2 which is now extending over two pages.
- Better alignment of the subtitles on the Figure 2.
- In the section "Risk score in the model with NTproBNP" in the methods section 1,75 % should be 1.75 % and 2,48 % should be 2.48 %
- Use of spaces between the number and following units should be controlled through the text of the manuscript.
ANSWER:
We have re-organised the figure 2, which reads clearer now. We have also revised the rest of the proposed typing queries along the manuscript.
REVIEWER 3
In the discussion relevance of the therapy aimed at the support of independent daily activities can be emphasized.
ANSWER:
According to the reviewer we have added this sentence to the discussion:
“....It is crucial to prescribe therapies to alleviate symptoms and anxiety at the various stages of the disease, as well as those treatments oriented to improve daily activities...”